# Effect of Heat Treatment on Microstructure and Properties of Clad Plates 316L/Q370qE

**DOI:** 10.3390/ma12091556

**Published:** 2019-05-12

**Authors:** Zimeng Wang, Shumei Kang, Meiling Xu, Yanqiang Cheng, Ming Dong

**Affiliations:** School of Material and Metallurgy University of Science and Technology LiaoNing, Anshan 114051, China; wzmustl@163.com (Z.W.); xumeilingustl@163.com (M.X.); 13029378215@163.com (Y.C.); dongmingustl@163.com (M.D.)

**Keywords:** clad plate, microstructure, interface, shear strength, neutral salt spray

## Abstract

Mechanical properties of Q370qE carbon steel can be improved by cladding it with 316L stainless steel. After rolling these materials together, microstructure, hardness, shear strength, and corrosion properties of the cladded metals were evaluated. Hardness and shear tests were performed according to appropriate standards to evaluate the bonds. The results show that the remarkable diffusion of Ni and Cr formed a hard transition zone. The width of this diffusion zone increases with increasing temperature. The shear strength of the clad interface reached a minimum of 385 MPa. In addition, the surfaces of samples were examined by neutral salt spray (NSS) tests and potentiodynamic polarization tests to examine corrosion behavior. The 316L side primarily exhibited pitting corrosion, while the Q370qE side was dominated by uniform corrosion.

## 1. Introduction

Roll-bonded cladding of metals produces large bonding areas between metal plates. Our stainless clad plate is made of cladding 316L stainless steel and Q370qE carbon steel. It is less costly than using the cladding layer alone (316L) and is widely used in chemical, petrochemical, and marine applications for its good corrosion resistance and mechanical properties [1,2]. There are many methods for producing stainless steel clad plate, among which, hot roll-cladding is the most economical and efficient manufacturing process. Kang et al. [3] believed that roll bonding was a solid-state welding process that connects similar or dissimilar metals. It is a mature and widely used manufacturing technique. For stainless steel clad plates, the performance is determined by both stainless steel and carbon steel, and the elemental and structural changes of the two metals influence the subsequent processing properties of the material. Rao et al. [4] studied the 304 stainless steel clad plate and observed the highest hardness at the interface because of the diffusion of chromium and nickel. Li et al. [5] studied the amount of oxides at the interface of stainless steel clad plates at different degrees of vacuum, and the results showed that the surface oxidation was more obvious at lower vacuum. Peng et al. [6] found that the fracture of oxides during the rolling process achieved a strong metallurgical bond between the metals on both sides of the interface, with finer oxides being more favorable for interface bonding. Although metallurgical and mechanical properties are heavily influenced by interface characteristics, there is little information in the literature on the bimetallic interface produced by cladding welding processes.

In this study, Q370qE is used as the base material and 316L is the cladding material. Because metallurgical and mechanical properties are greatly influenced by interface characteristics, the current research work is to study the interdiffusion phenomenon and interface microstructure after heat treatment at 500 °C, 600 °C, 800 °C and 1000 °C. 

## 2. Experiments

As mentioned above, the flyer plate is 316L and the parent plate is Q370qE. A schematic representation of the process for 316L/Q370qE clad plate is shown in Figure 1. It includes three steps: surface cleaning, assembly, heating and rolling. The primary chemical compositions of 316L and Q370qE are given in Table 1. The samples were heat-treated at 500 °C, 600 °C, 800 °C and 1000 °C for 30 min.

The test samples had a cross-sectional area of approximately 100 mm^2^ (10 mm × 10 mm). Both surfaces were etched: Q370 with 4% nitric acid alcohol solution and 316L with aqua regia. After the tests, the surfaces of the samples were examined using a scanning electron microscope (SEM, Carl Zesis AG, Jena, Germany) and subjected to mechanical and electrochemical testing. Vickers hardness tests were performed on a Huayin310HVS-5-type Vickers hardness machine (Laizhou Huayin Testing Instrument Co., Ltd., Laizhou, China) with a 100 g load for 10 s. The shear test was carried out on a Universal Testing Machine with reference to the Chinese standard GB/T 6396-2008. The cladding plates were subjected to neutral salt spray tests with 5% solution of NaCl in distilled water at room temperature. Polarization tests were performed in a 3.5 pct NaCl solution at room temperature using a CS350 electrochemical workstation (Corrtest, Wuhan, China) with a three-electrode device consisting of a silver chloride reference electrode, a platinum auxiliary electrode, and the samples. The scanning potential was in the range of −1~0.5 V, and the scan rate was 0.05 mV/s.

## 3. Results and Discussion

### 3.1. Metallographic Examination

It can be seen from Figure 2 that the Q370qE microstructure is mainly strip-like ferrite and lamellar pearlite. As the heat treatment temperature is increased from 500 °C to 1000 °C, the grain volume near the interface becomes larger. This is because grains at the interface are severely deformed during the rolling process and subsequently recrystallized during heat treatment. As shown in Figure 1a–c, this region is referred to as the full ferrite region or semi-decarburized region because no pearlite is present [7]. Since the mass fraction of the carbon steel side C is 0.10%, the mass fraction of the stainless steel side C is 0.02%, the C content on both sides of the composite interface has a higher concentration gradient, and the carbon atom size is also fine, which is a typical gap diffusion and diffusion speed fast. The pearlite grains form a decarburized layer near the composite interface. As the temperature increases, the width of the decarburized layer becomes wider. The heat treatment temperature of the composite plate increases, and the C atoms belong to the gap diffusion, so the distant C atoms diffuse to the composite interface, thereby increasing the width of the decarburization layer.

It can be seen from Figure 3 that the microstructure of the 316L side is austenite with clear grain boundaries and substantial twinning. The grain boundary at 800 °C is the most obvious. Because of the heat treatment at 800 °C, carbon and chromium easily form chromium carbides, reducing the chromium content at the grain boundary and forming a chromium-depleted zone [8,9]. Heat treatment causes the carbon and alloying elements in the 316L stainless steel and the carbon steel Q370qE to diffuse to the grain boundaries, thereby causing more carbide particles at the grain boundaries. This can seriously affect the performance of the composite sheet.

Figure 4a–f shows the diffusion curves of Fe, Ni, and Cr at the bonding interface of the composite plate after heat treatment at 500 °C, 600 °C, 800 °C, and 1000 °C, respectively. Because 316L stainless steel contains more chromium and nickel, while carbon steel Q370qE contains more iron, at high temperatures, chrome and nickel diffuse into Q370qE and carbon diffuses into 316L. Obviously, the diffusion of elements such as Cr and Ni is not conducive to the corrosion resistance of the stainless steel side. It can be seen from the figure that a diffusion region appears at the interface. At 500 °C, 600 °C, 800 °C, and 1000 °C, the diffusion regions were 4.05 μm, 4.12 μm, 8.90 μm, and 10.61 μm thick, respectively. It can be concluded that as the temperature increases, the degree of elemental diffusion also increases.

### 3.2. Microhardness Tests

Hardness and shear tests were used to evaluate quality in this study. The cladded plates were harder than either the 316L or Q370qE. The hardness distribution at the interface is shown in Figure 5. The cladding plate treated at 800 °C has the highest hardness and the best performance. This is because carbon and chromium easily form chromium carbide at 800 °C heat treatment. As demonstrated in Figure 1, combined with the results of the microstructure analysis of the composite interface and the element distribution, it can be seen that the change in hardness is consistent with the decarburization layer and the diffusion layer of the base layer. A peak is reached at the boundary, indicating that at this boundary, the proportion of each element is more likely to form carbides. The hardness near the interface on the Q370qE side is low because the carbon diffuses to the 316L side, forming a decarburized layer consisting primarily of low hardness ferrite. This is confirmed with Figure 2. When the diffusion layer was measured, the hardness sharply increased and reached a peak at the interface, and the hardness began to decrease after leaving the interface. This can be seen in conjunction with Figure 5, because the Cr and Ni in the complex diffuse to the composite interface due to the concentration gradient and react with the diffused C in the carbon steel to form carbides, thereby increasing the hardness.

The shear tests were carried out in accordance with the Chinese standard GB/T 6396-2008, which states that the shear strength of the cladding plates shall not be less than 210 MPa. As described elsewhere [10], the shear strength of the cladded joint should be higher than that of either component, the shear strength of 316L/Q370qE cladding above 370 MPa. Figure 6 shows that the shear strength of the samples was greater than both the national standard of 210 MPa and 350 MPa. This indicates that the interface bonding effect is good.

### 3.3. Neutral Salt Spray Tests

The clad metal was subjected to a neutral salt spray test in a 5% NaCl solution to check its corrosion behavior. As shown in Figure 7, Q370qE suffers severe corrosion, and the gap trench and etch pit are clearly visible. In contrast, the corrosion rate of 316L stainless steel is much lower than that of the Q370qE. In the 5% NaCl solution, since the volume of Cl^−^ is fine, it can cause corrosion. However, Mo is added to 316L, which together with Cr constitutes a higher density oxide film, so that Cl^−^ is difficult to enter. Thus, 316L corrosion resistance is very good. Figure 7 shows that etching causes the most severe corrosion in cladded plates heat treated at 800 °C, as these composite sheets appear to be sensitized. 

### 3.4. Potentiodynamic polarization tests

Figure 8 and Table 2 show the results of the polarization tests of the samples in 3.5% NaCl solution. An evaluation of the potentiodynamic test results showed that the cladding composite treated at 500 °C had the lowest I_corr_ and therefore the best corrosion resistance. Since the sample after 800 °C heat treatment is sensitized, the precipitated phases in the grain boundary increase the current and corrosion tendency. Thus, the sample after heat treatment at 800 °C has the worst corrosion resistance.

## 4. Conclusions

(1) In the 316L/Q370qE composite plate structure, the Q370qE was predominately ferrite with a refined grain structure. Decarburization occurs near the interface. The 316L is solely austenite with a uniform grain size. At the interface of the composite plates, a diffusion region appears in which chromium carbide is precipitated.

(2) The composite plates are well bonded at the interface. The mechanical properties of the composite plate are improved due to grain refinement, and diffusion improves the bonding performance. The hardness of 316L increased in proximity to the joint, and the hardness of Q370qE decreased on approaching the joint. When the heat treatment temperature was 500 °C, the shear strength was the highest and the interface was the strongest. The interface shear strength decreased with rise of heat treatment temperature.

(3) Neutral salt spray tests and polarization tests show that when the heat treatment temperature was 800 °C, the chromium content near the grain boundary and the corrosion resistance decreases. The corrosion resistance of the composite plate is worst after heat treatment at 800 °C.

## Figures and Tables

**Figure 1 materials-12-01556-f001:**
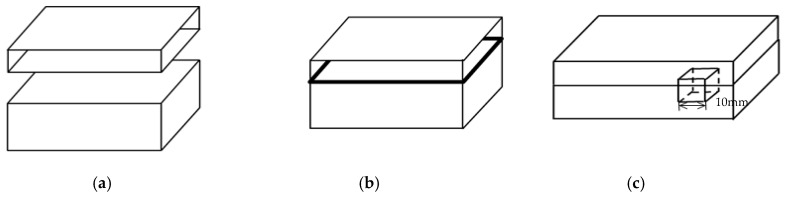
Schematic representation of the process for 316L/Q370qE clad plate (**a**) surface treatment and stacking; (**b**) hot rolling; (**c**) sampling schematic.

**Figure 2 materials-12-01556-f002:**
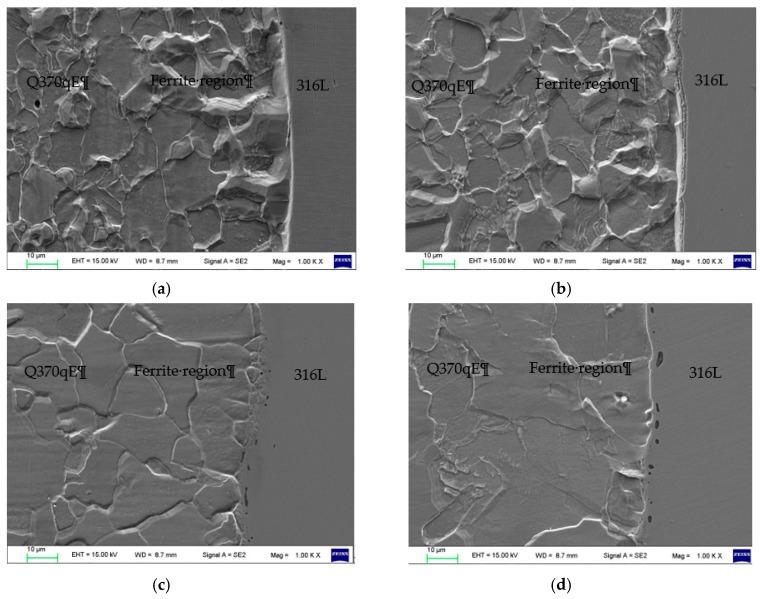
Microstructures of the Q370qE side (**a**) 500 °C; (**b**) 600 °C; (**c**) 800 °C; (**d**) 1000 °C.

**Figure 3 materials-12-01556-f003:**
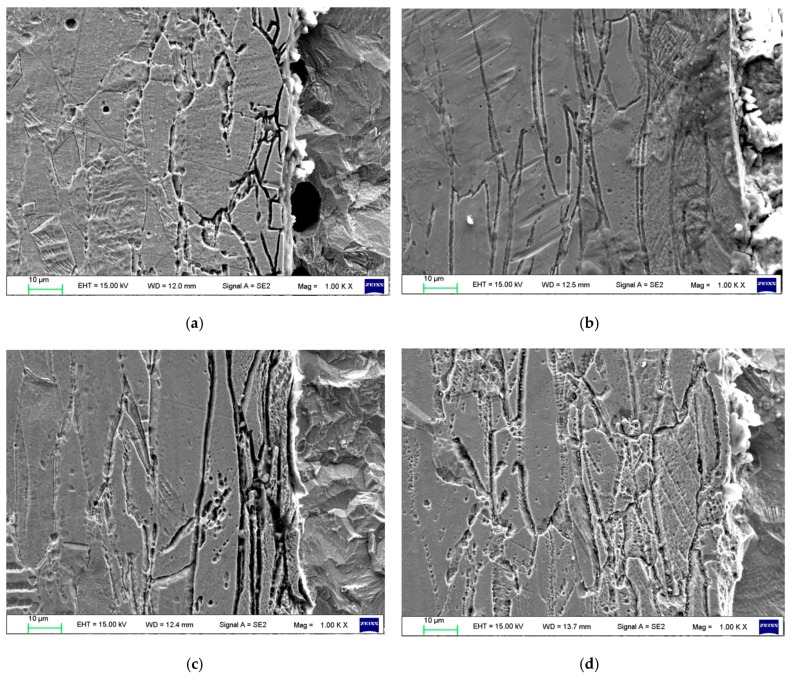
Microstructures of the 316L side and SEM results: (**a**) 500 °C; (**b**) 600 °C; (**c**) 800 °C; (**d**) 1000 °C.

**Figure 4 materials-12-01556-f004:**
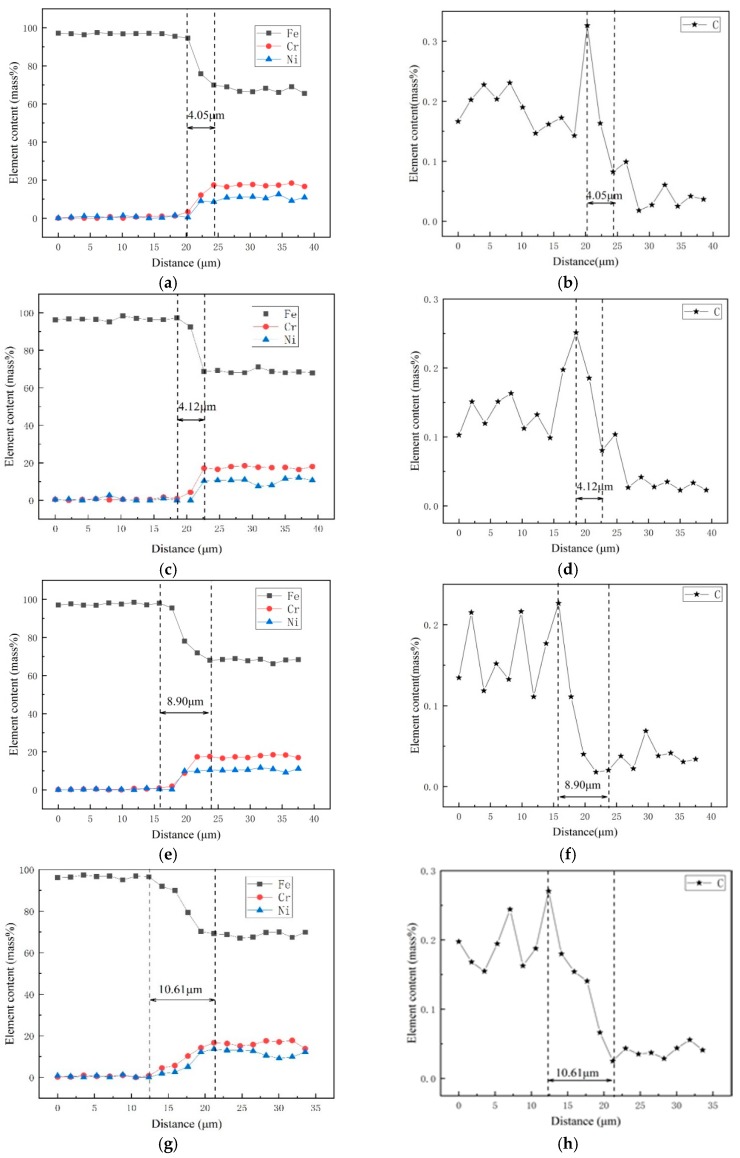
Line analyses of transition zone: (**a**,**b**) 500 °C; (**c**,**d**) 600 °C; (**e**,**f**) 800 °C; (**g**,**h**) 1000 °C.

**Figure 5 materials-12-01556-f005:**
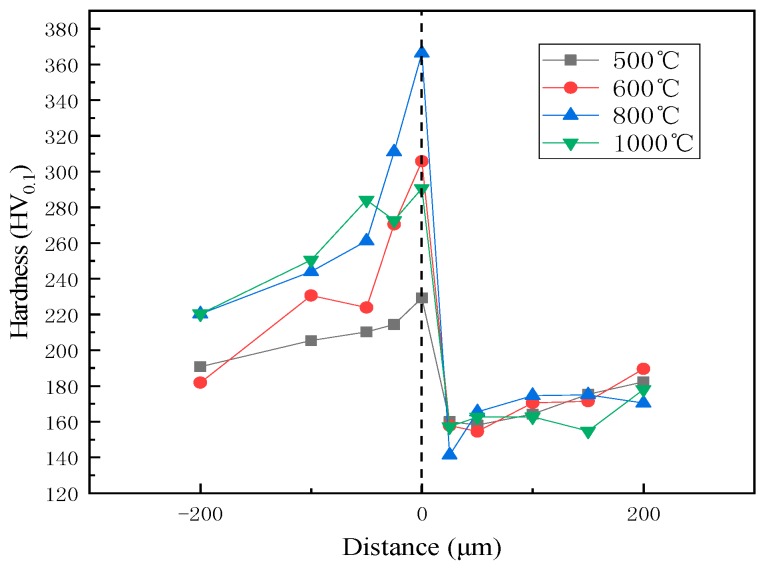
Hardness distribution near phase interface.

**Figure 6 materials-12-01556-f006:**
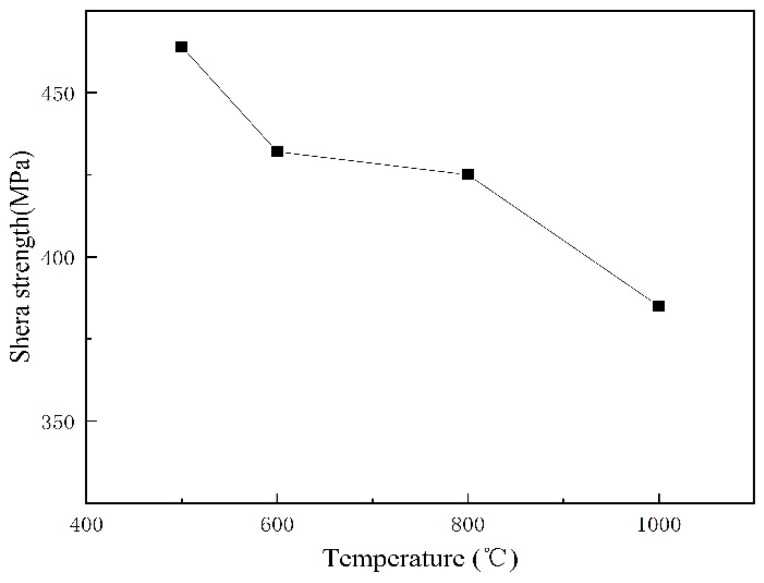
Shear strength of the interface.

**Figure 7 materials-12-01556-f007:**
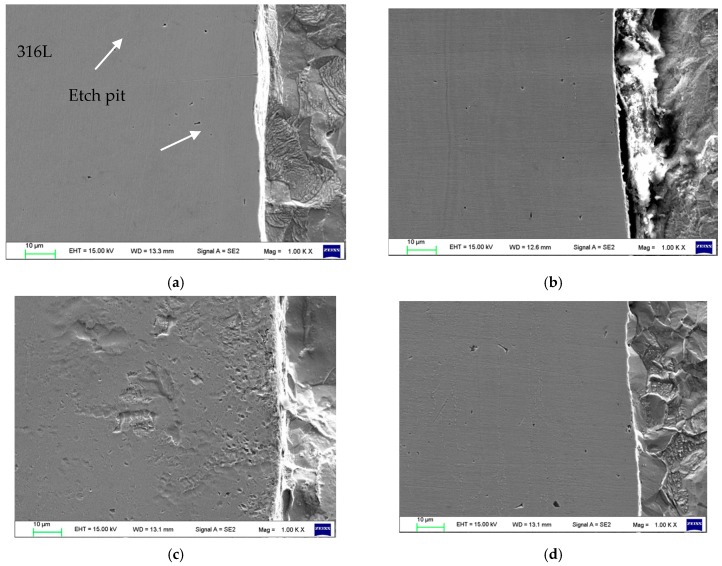
Microstructures of the interface: (**a**) 500 °C; (**b**) 600 °C; (**c**) 800 °C; (**d**) 1000 °C.

**Figure 8 materials-12-01556-f008:**
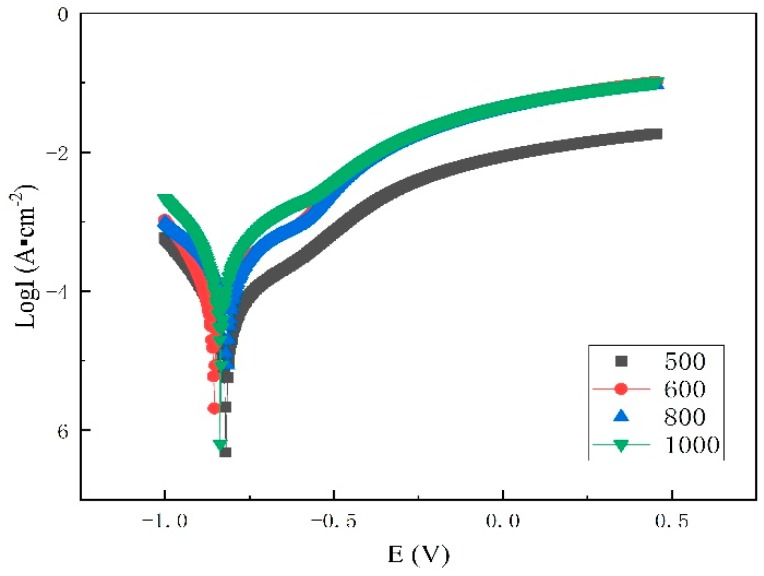
Polarization curve at the interface.

**Table 1 materials-12-01556-t001:** Chemical composition of 316L/ Q370qE steel (wt %).

Elements	C	Si	Mn	S	P	Cr	Ni	Mo	Fe
316L	0.02	0.56	1.33	0.002	0.031	17	10	2	balance
Q370qE	0.10	0.20	1.55	0.003	0.015	–	–	–	balance

**Table 2 materials-12-01556-t002:** Corrosion potential and current at the interface.

t/°C	500	600	800	1000
E/V	−0.82	−0.85	−0.81	−0.837
I/A cm^−2^	4.79 × 10^−7^	2.07 × 10^−6^	8.65 × 10^−6^	6.41 × 10^−7^

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
