# Peer review of "Effect of Heat Treatment on Microstructure and Properties of Clad Plates 316L/Q370qE"

_materials, 2019, doi:10.3390/ma12091556_

Reviewer 1 Report

The authors of the reviewed manuscript performed complex characterization of the microstructure, mechanical properties and corrosion resistance of the 316L/Q370qE clad plates after heat treatment. The presented results are very important not only from the scientific point of view but also from the practical application point of view. It is noteworthy, that the experiment was well designed and the results were clearly described. Unfortunately, in the part Results and discussion the Authors describe the individual obtained results, however, any interpretation and scientific discussion of them with the literature data is given. This is the most important reason for which the manuscript paper needs to be rewritten and Authors having the required information should try to make more in-depth discussion and not just to show the experimental analyzed results.

Summarizing, before the publication of this paper the Authors must put some effort referring to the important issues listed above.

Author Response

Response to Reviewer 1 Comments

Point 1 In the part Results and discussion the Authors describe the individual obtained results, however, any interpretation and scientific discussion of them with the literature data is given.

Response 1:

(1) In the results and discussion, the influence of heat treatment on the microstructure of stainless steel clad plate was analyzed in more detail. The influences of microstructure on mechanical properties and corrosion resistance were combined.

(2) The diffusion of C element in heat treatment was explained in depth. And the reason why the decarburized layer widens as the temperature increases was added in the paper.

(3) Explain the reason why the hardness changes on the carbon steel side and the stainless steel side, respectively. Decarburization layer forms on the carbon steel side. Carbides formed at the interface such that the hardness at the interface peaks. The Cr and Ni in the stainless steel form carbides with the diffused C, so that the hardness near the interface is relatively high.

(4) In the neutral salt spray experiment, the effect of Cl- on corrosion rate was analyzed. The corrosion formation mechanism and influencing factors of stainless steel clad plate were studied.

Thank you for your review.

Reviewer 2 Report

The introduction is quite general, and it falls in short explaining of literature pertaining the specific research domain.

In Experiments section, it should be a deeper explanation of how the roll bonded cladding processes was done for this study.

In Table 1, should be included the error of the chemical composition analysis.

It should be done a schematic figure in order to explain from where the different test samples were obtained.

Author Response

Response to Reviewer 2 Comments

Point 1: The introduction is quite general, and it falls in short explaining of literature pertaining the specific research domain.

Response 1The current research developments of stainless steel clad plate were added in the introduction. References were revised. Literatures related to hot-rolled composite panels were added to make the introduction part more consistent with the article.

Point 2:  In Experiments section, it should be a deeper explanation of how the roll bonded cladding processes was done for this study.

Response 2Figure 1(a)(b) explains the hot rolling process. The rolling steps are briefly summarized in the paper.

Point 3: It should be done a schematic figure in order to explain from where the different test samples were obtained

Response 3Figure 1(c) shows the sampling at the interface of stainless steel clad plates.

Point 4:  In Table 1, should be included the error of the chemical composition analysis.

Response 4Specification of tabular data by consulting relevant literature and modification of Table 1.

Thank you for your review.

Round  2

Reviewer 1 Report

The manuscript seems to me of good standard after the comments of the referees were taken into account. Upon reading the revised version, I recommend the manuscript for publication in current state.